# On the Strong Correlation Between Model Invariance and Generalization

**Weijian Deng    Stephen Gould    Liang Zheng**
Australian National University
`{firstname.lastname}@anu.edu.au`

## Abstract

Generalization and invariance are two essential properties of machine learning models. Generalization captures a model's ability to classify unseen data while invariance measures the consistency of model predictions on transformed data. Existing research suggests a positive relationship: a model generalizing well should be invariant to certain visual factors. Building on this qualitative implication we make two contributions. First, we introduce effective invariance (EI), a simple and reasonable measure of model invariance which does not rely on image labels. Given predictions on a test image and its transformed version, EI measures how well the predictions agree and with what level of confidence. Second, using invariance scores computed by EI, we perform large-scale quantitative correlation studies between generalization and invariance, focusing on rotation and grayscale transformations. From a model-centric view, we observe generalization and invariance of different models exhibit *a strong linear relationship*, on both in-distribution and out-of-distribution datasets. From a dataset-centric view, we find a certain model's accuracy and invariance *linearly correlated* on different test sets. Apart from these major findings, other minor but interesting insights are also discussed.

## 1   Introduction

Generalization and invariance are two important model properties in machine learning. The former characterizes how well a model performs when encountering in-distribution or out-of-distribution (OOD) test data [1–5]. The latter assesses how consistent model predictions are on transformed test data [6–12]. Therefore, understanding how the two properties are related would benefit model decision analysis under dynamic environments.

The importance of model invariance on generalization has been *qualitatively* explored [9, 12–15]. For example, adding rotation invariance to the model improves its in-distribution (ID) classification accuracy [14, 16, 17]; a shift-invariant model is robust to perturbation [9]. In addition, some works provide a neuroscientific perspective to investigate the importance of invariant representations for pattern recognition [13, 15, 18–22]. Furthermore, theoretical investigations suggest that learning invariant features benefits model generalization [12, 22]. However, most existing research is limited to a few ID datasets and model architectures. As such, the relationship of interest remains unknown in many other scenarios, such as OOD and large-scale test datasets, and other types of models.

Before performing the quantitative study, it is necessary to first quantify generalization and invariance. For the former, the deep models we consider are well trained, so we simply use the accuracy on the test set, as in many previous works [3, 2, 23]. In comparison, quantifying invariance is not as straightforward. Some works use model accuracy drop when the test set undergoes transformations to indicate invariance ability [24, 7, 17]. While this strategy is useful for a single model, its effectiveness is limited when comparing the invariance of multiple models. Others resort to consistency, *i.e.*,

36th Conference on Neural Information Processing Systems (NeurIPS 2022).

models should have the same decision [9, 6], but this method neglects prediction *confidence*, which we find critical for describing invariance (see discussions in Section 3).

We make two contributions to the community. **First**, we propose a new method to measure model invariance, named effective invariance (EI), which considers both the consistency and confidence of predictions. Given a test image and its transformed counterpart, if the model predicts the same class with high confidence, the EI value or invariance strength is high. Otherwise, if the model makes different class predictions or the confidence is low, the EI score will be low. We show this new measure solves invariance valuation in canonical cases where the commonly used metrics (*e.g.*, Jensen-Shannon divergence) may fail. **Second**, we conduct a broad correlation study to quantitatively understand the relationship between model generalization and invariance. Specifically, we use 8 test sets with various distribution types, such as the in-distribution ImageNet validation set [25], and out-of-distribution ImageNet-Rendition with style shift [4]. We evaluate 150 ImageNet models ranging from traditional convolution neural networks (*VGGs* [26]) to the very recent vision transformers (*e.g.*, BEiT [27]). Below we list two key observations and example insights.

- For *various models*, there is a strong correlation between their accuracy and invariance on both in-distribution and out-of-distribution datasets (Sections 5.1 and 5.2). This finding can be useful for unsupervised model selection because EI does not require test ground truths.
- On *various out-of-distribution datasets*, a model's accuracy and EI scores are also strongly correlated (Section 6). This observation can be used to predict model accuracy on out-of-distribution datasets without access to ground truths.
- Compared with data augmentation, training with more data seems more effective to improve invariance and generalization (Section 5.6).

## 2  Related Work

**Predicting generalization gap: a model-centric view.** This task aims to predict the generalization gap of machine learning models on in-distribution data, *i.e.*, the difference between training and test accuracy. Most existing works focus on developing *complexity measure* of trained network parameters and training data [28–36], such as persistent topology [31] and the product of norms of the weights across layers [33]. These methods, assuming the training and test distributions are the same, *do not* consider the characteristics of the test distribution. Moreover, they only study limited types of neural networks. In comparison, we conduct a much more comprehensive study on both in-distribution and out-of-distribution test sets, using various network architectures. We show that invariance, under our definition, serves as a strong indicator of model generalization ability or accuracy on both in-distribution and out-of-distribution test data.

Closely related to our work, two recent methods [37, 24] predict ID generalization gap based on how a network performs on perturbed data points. Specifically, Kashyap *et al.* [37] use confidence drop to represent invariance, which is less effective for invariance measurement under OOD data. Schiff *et al.* [24] uses accuracy drop to measure invariance, which needs test labels, while EI does not require test labels and is more reasonable than accuracy under OOD data. In addition, drawing a response curve in [24] is computationally heavy, while our method is relatively efficient. Further, both studies are limited in their scope: they mainly study ID generalization, has few types of networks and lack large-scale test sets, while our work is much more comprehensive.

**Predicting generalization gap: a dataset-centric view.** The overall goal of this task is to predict the performance of a given model on various unlabeled test sets [38–43]. Many methods take into account the statistics of the test set for accuracy prediction [38, 39, 44, 40, 45], such as distribution shift [38], average Softmax score on each test set [39]. We contribute a new solution: using the model's invariance on the OOD dataset to predict its accuracy. This is supported by our new observation of a strong linear correlation between a certain model's accuracy and invariance on various test sets.

**Improving robustness with data augmentation.** Data augmentation transforms training data to increase its diversity, which helps learn more robust models [46–52]. For example, Mixup [53, 54] and AutoAugment [48] are shown to improve model performance under distribution changes [55, 51].

Instead of using common transformations, adversarial training [56–58] augments training images with an adversarially learned noise distribution. While these works aim to improve corruption robustness with data augmentation, we instead use the latter to analyze model invariance (and generalization).

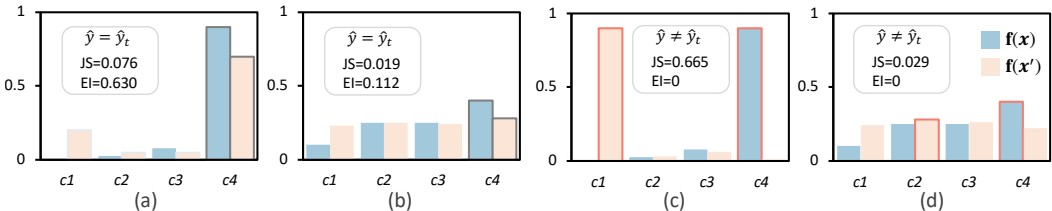

Figure 1: **An illustrative comparison of EI and Jensen-Shannon divergence (JS) as invariance measures.** Four representative cases are shown, where for each class ($c1$-$c4$) we show the Softmax output of the original image $\boldsymbol{f}(\boldsymbol{x})$ and the transformed image $\boldsymbol{f}(\boldsymbol{x'})$. On the one hand, the model exhibits higher invariance in **(a)** than **(b)**, because it makes the same class predictions ($\hat{y} = \hat{y}_t$) and has higher confidence in (a). This is correctly reflected by EI (0.630 *vs.* 0.112; higher is better), but JS incorrectly decides the opposite way (0.076 *vs.* 0.019; lower is better), because JS does not consider confidence explicitly. In cases **(c)** and **(d)**, the model makes different class predictions ($\hat{y} \neq \hat{y}_t$), so its invariance should be very low. This is again correctly captured by EI (0 value for both cases), but JS erroneously gives high invariance to (d), due to the fact that JS merely looks at the global shape of the Softmax vectors without explicitly considering class prediction consistency.

## 3 Proposed Effective Invariance (EI)

**Notations**. Considering an $K$-way classification task, we define input space $\mathcal{X} \in \mathbb{R}^d$ and label space $\mathcal{Y} = \{1, ..., K\}$. Given a sample $(\boldsymbol{x}, y)$ drawn from an unknown distribution $\pi$ on $\mathcal{X} \times \mathcal{Y}$, a neural network classifier $\boldsymbol{f} : \mathbb{R}^d \to \Delta_K$ produces a probability distribution for $\boldsymbol{x}$ on $K$ classes, where $\Delta_K$ denotes the $K - 1$ dimensional unit simplex. Specifically, $f_i(\boldsymbol{x})$ denotes the $i$-th element of the Softmax output vector produced by $\boldsymbol{f}$. Then, $\hat{y} =: \arg\max_i f_i(\boldsymbol{x})$ is the predicted class, and $\hat{p} =: \max_i f_i(\boldsymbol{x})$ is the associated confidence score. Image transformation is defined as $\mathcal{T} : \mathbb{R}^d \to \mathbb{R}^d$. Then, the transformed image is $\boldsymbol{x'} = \mathcal{T}(\boldsymbol{x})$, and its predicted class is $\hat{y}_t =: \arg\max_i f_i(\boldsymbol{x'})$ with confidence score $\hat{p}_t =: \max_i f_i(\boldsymbol{x'})$.

**Drawback of existing invariance measures**. A commonly seen strategy is to directly use the distance between the Softmax vectors of two predictions as an invariance measure: a lower distance means higher invariance, and vice versa. Examples of the similarity metrics are Jensen-Shannon divergence (JS) [59, 46] and $\ell_2$ distance [60, 61] and Kullback–Leibler divergence [62, 63]. However, they only leverage the global similarity between two Softmax vectors without explicit consideration of prediction class consistency and confidence. We illustrate this drawback by taking JS divergence as an example in Fig. 1. In cases (a) and (b) where the predicted classes are both consistent, JS decides classifier $\boldsymbol{f}$ in (b) has higher, which ignores the low confidence in (b). In cases (d) where the predicted classes are different, JS still gives high invariance (small JS score), indicating a clear error. Moreover, in works [46, 60, 61] cases (b) and (d) are discarded when computing the consistency loss.

**Definition of EI.** Unlike neuroscience works that study the invariance of an individual neuron of the network [13, 15, 18, 19], we measure the invariance at the network level. Specifically, given an image and its transformed sample, a model with high invariance should give the same predicted class, and vice versa [6, 9]. In our definition of EI, we further use prediction confidence. Our motivation is as follows. When a model predicts the same class for the two images, if either of the two predictions is of low confidence, we should not consider it as highly invariant but give a penalty. A model should have a high invariance if and only if it is highly confident in predicting the same class. Based on these considerations, EI is defined as:

$$\text{EI}(\boldsymbol{x}, \mathcal{T}(\boldsymbol{x}), \boldsymbol{f}) = \begin{cases} \sqrt{\hat{p}_t \cdot \hat{p}} & \text{if } \hat{y}_t = \hat{y}; \\ 0 & \text{otherwise}. \end{cases} \tag{1}$$

To better understand the soundness of EI, we depict four representative cases in Fig. 1. In cases (a) and (b), classifier $\boldsymbol{f}$ gives the same predicted class ($\hat{y} = \hat{y}_t$) on original and transformed images. Under EI, Classifier $\boldsymbol{f}$ has a higher invariance ability in (a) because it has high confidence scores. In case (c), the predicted class are different ($\hat{y} \neq \hat{y}_t$), and in (d), the predictions are low in confidence (and give different classes), so we define invariance in both cases to be 0.

**Computation of EI in practice.** Given a test image, we generate a transformed image using a certain transformation. Then, we compute the EI score based on their Softmax vectors (Eq. 3). We obtain the model invariance by averaging the EI scores over all the test images. In this work, we mainly investigate the rotation transformation and grayscale transformation. For the former, to avoid interpolation that would introduce artifact, we only use three transformation angles $(90°, 180°, 270°)$. For each rotation angle, we compute an overall invariance score by comparing it with the predictions of the original data. By averaging the three EI scores, we obtain rotation invariance on the test set. For the grayscale transformation, we remove color information and keep only luminous intensity information. Then, we compare the predictions of grayscale and original data and compute the overall grayscale invariance on each test set.

## 4 Experimental Setup

### 4.1 Models to Be Evaluated

We consider both very recent and classic image classification models with different architectures, including Convolutional Neural Networks (*e.g.*, standard VGGs [26], ResNets [64], and modern ConvNeXt [65]), Vision Transformers (*e.g.*, ViTs [66], Swin [67], and BEiT [27]), and all-MLP architectures [68, 69] (*i.e*, MLP-Mixer [69]).

In addition to different architectures, we also cover models with various training and regularization strategies (*e.g.*, learning rate schedule [70], label smooth [71] and data augmentation [46, 47, 49, 48]), scaling strategies in model dimension (width, depth, and resolution) [72–74], and learning manners (supervised learning, semi-supervised learning [75] and knowledge distillation [76, 77]). In total, we have **150 models** provided by TIMM [78]. They are either trained or fine-tuned on the ImageNet-1k training set [25]. The selected models can be roughly divided into the following three categories:

**Standard neural networks.** This category includes 100 models only trained on ImageNet training set. These networks cover various architectures ranging from VGGs [26] to EfficientNet [73].

**Semi-supervised learning.** We include 15 models trained in a semi-supervised learning manner. They leverage a large collection of unlabelled images of YFCC100M [79] or Instagram 900M [80] to improve the performance. We use models trained based on a teacher-student paradigm (*e.g.*, SWSL-ResNet [75] and SSL-ResNet [75]). Models trained with self-training methods on unlabeled JFT-300M [81] (*e.g*, EfficientNet-L2-NS [82]) are also included.

**Pretraining on more data.** We use another 35 models that are pre-trained on significantly larger datasets than the standard ImageNet training set. Specifically, we consider three pre-training methods: a) weakly supervised pretraining on *IG-3.6B* (*i.e.*, RegNetY [83] and ResNeXt101-WSL [80]); b) supervised pre-training on ImageNet-21K [25] (*e.g.*, BiT [84] and Swin [67]); (c) supervised pretraining on JFT-300M [81] (*e.g.*, ViT L/16 [66]).

### 4.2 Test Sets

We use both in-distribution (ID) and out-of-distribution (OOD) datasets for the correlation study. Specifically, the ImageNet validation set (ImageNet-Val) is used as ID test set. For OOD test sets, we use seven datasets, each with a different distribution from standard ImageNet. Their distribution shift can be divided into the following five types.

**Dataset reproduction shift.** ImageNet-V2 [23] is a recollected version of ImageNet-Val. It contains three versions resulting from different data sampling strategies: Matched-Frequency (A), Threshold-0.7 (B), and Top-Images (C). Each version has $10,000$ images from $1,000$ classes.
**Natural adversarial shift.** ImageNet-Adv(ersarial) [85] is adversarially selected to be misclassified by ResNet-50. Its natural adversarial examples are unmodified real-world images and have been shown to be hard for other models [85, 3]. It has $7,500$ samples from 200 ImageNet classes.
**Sketch shift.** ImageNet-S(ketch) [86] consists of sketch-like images and matches ImageNet-Val in categories and scale. It contains $50,000$ images and shares the same $1,000$ classes as ImageNet.
**Blur shift.** We use ImageNet-Blur with the highest severity provided by [87]. This dataset is synthesized by blurring ImageNet-Val images with a Gaussian function.
**Style shift.** ImageNet-R(endition) [4] contains various abstract visual renditions (*e.g.*, art, paintings, and video game) of ImageNet classes. ImageNet-R has $30,000$ images of 200 ImageNet classes.

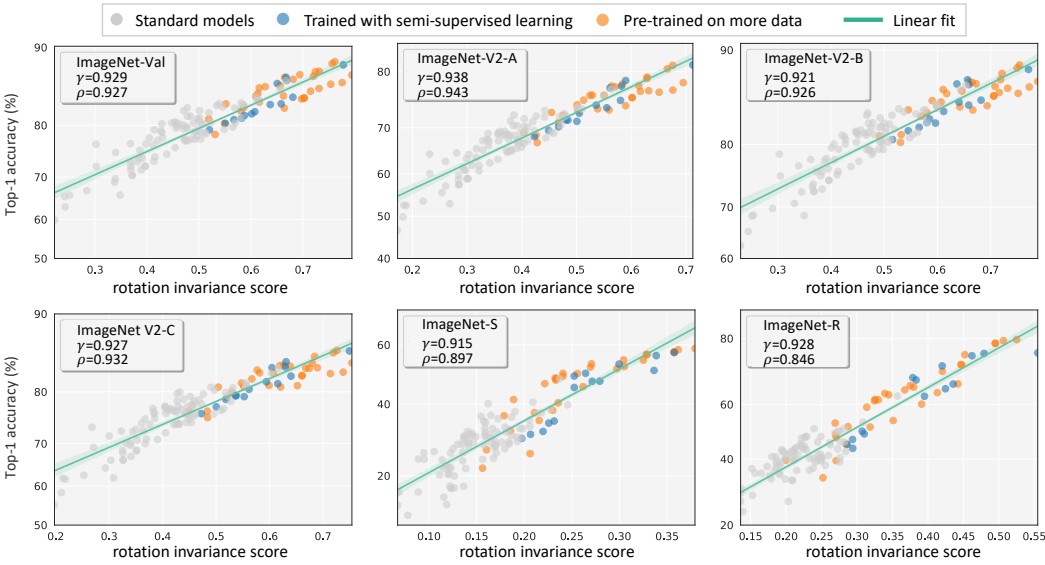

Figure 2: **Correlation between accuracy (%) and rotation invariance (EI) for** 150 **models.** Each figure is obtained from testing on a different ImageNet test set. In each figure, each dot denotes a model, and straight lines are fit by robust linear fit [90]. The shaded region in each figure is a 95% confidence region for the linear fit from 1,000 bootstrap samples. We clearly observe a strong linear relationship (Pearson's Correlation $r > 0.915$ and Spearman's Correlation $\rho > 0.875$).

## 4.3 Correlation Measures

We use Pearson Correlation coefficient ($r$) [88] and Spearman's Rank Correlation coefficient ($\rho$) [89] to measure the linearity and monotonicity between invariance and generalization, respectively. Both coefficients range from $[-1, 1]$. A value closer to $-1$ or $1$ indicates a strong negative or positive correlation, respectively, and $0$ implies no correlation [88]. To precisely show the correlation, we use logit axis scaling that maps the accuracy range from $[0, 1]$ to $[-\infty, +\infty]$, following [3, 2]. Unless noted otherwise, the correlation coefficients are calculated using invariance score and non-linearly scaled accuracy numbers.

## 5 Experimental Observations

The experiment is from a model-centric perspective, where we investigate how different models' accuracy and invariance correlate, and a series of observations are made (Section 5.1 - Section 5.6).

## 5.1 Strong Correlation Between Model's Rotation Invariance and Accuracy

In Fig. 2, we show the correlation results of rotation invariance and generalization. We have two observations. **First**, we find that for different models, their rotation EI scores have a linear relationship with their classification accuracy. The correlation holds for both ID test and OOD test sets, various architectures, and training strategies. Specifically, both correlation metrics $\lambda$ and $\rho$ are higher than 0.840. This indicates models with higher accuracy numbers are most likely to have stronger rotation invariance (measured by EI), and vice versa. To our knowledge, it is a very early observation of the quantitative relationship between generalization and invariance (to a certain factor).

**Second**, training with more data benefits rotation invariance and generalization. Large datasets contain images with various geometric variations. When (pre)trained with large datasets, models (blue and orange dots in Fig. 2) adapt to the rotation variations and gains stronger invariance, which, according to our study, likely means a higher generalization accuracy on ID and OOD test sets.

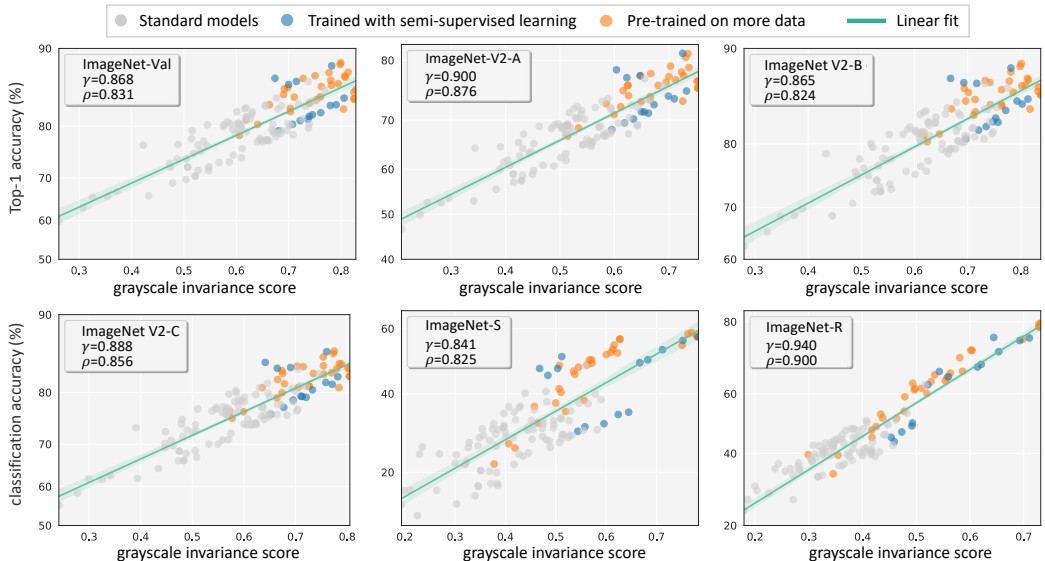

Figure 3: **Correlation between accuracy (%) and grayscale invariance (EI) of** $150$ **models.** Similar to Fig. 2, each dot denotes a model, where different colors denote different training strategies (see Section 4.1). The subfigures correspond to three ImageNet test sets, respectively. We observe the correlation is also strong: Pearson's Correlation $r > 0.840$, and Spearman's Correlation $\rho > 0.820$.

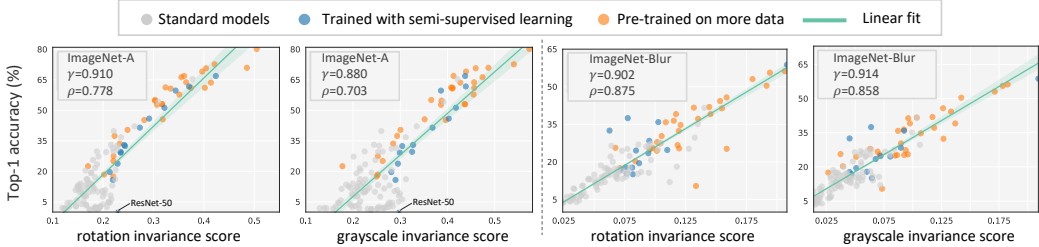

Figure 4: **Correlation study on very hard test sets.** We use the very hard ImageNet-A and Image-Blur for testing, where the accuracy of $83$ models is lower than $20\%$. Both rotation invariance and grayscale invariance are evaluated. Overall we observe a relatively solid correlation in all four cases. Looking more closely, most standard models (gray dots) are scattered in the low-accuracy region, while models trained with more data (blue and orange ones) move away from this region and exhibit linear trends. Thus, the overall linear correlation is high ($r > 0.880$), and the overall rank correlation is slightly less consistent ($\rho$ ranges from $0.703$ to $0.858$) but still has clear trends.

## 5.2 Strong Correlation Between Model's Grayscale Invariance and Accuracy

We now focus on grayscale invariance and report the correlation results in Fig. 3. We have the following conclusions. **First**, among the $150$ models, there is a strong linear correlation between accuracy and grayscale invariance measured by EI. Specifically, both correlation coefficients $r$ and $\rho$ are higher than $0.820$ on all test sets. **Second**, we find that pretrained or semi-supervised models tend to have higher grayscale invariance and accuracy than standard models, which again indicates the usefulness of large training sets. **Third** and interestingly, the correlation is stronger on ImageNet-R than ImageNet-Val and ImageNet-V2-A ($0.940$ *vs.* $0.900$ *vs.* $0.868$). In fact, ImageNet-R is featured by style shift during its collection [4], so for this test set being invariant to color changes is an important property for a stronger generalization ability.

## 5.3 Correlation Exists on Very Hard Test Sets

We now study the correlation under very hard test sets and use ImageNet-A and ImageNet-GaussBlur for testing, on which the accuracy of $83$ models is lower than $20\%$. We study rotation invariance and

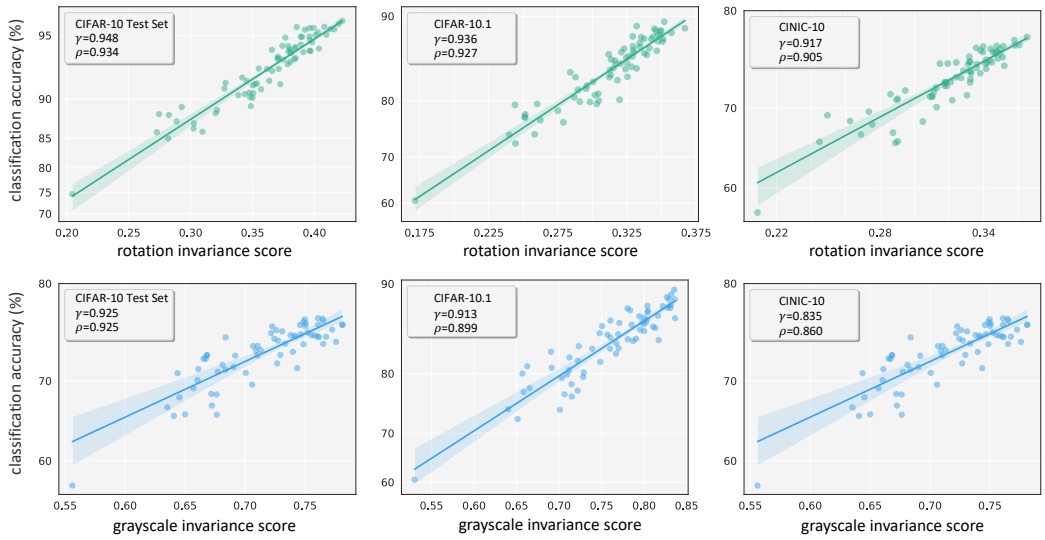

Figure 5: **Correlation study under the CIFAR-10 setup.** Each data point is a CIFAR model. We report the correlation results with rotation invariance (top) and grayscale invariance (bottom). We observe that a strong correlation exists between accuracy and invariance on all three test sets.

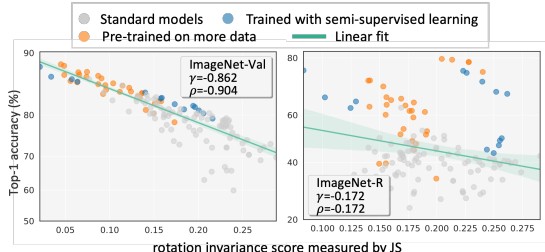

Figure 6: **Correlation between accuracy (%) and invariance (JS)**. Despite of the good correlation on ImageNet-Val, the correlation on ImageNet-R, a harder test set, is weak. More results and analysis are provided in the supplementary materials.

| Test set | JS | EI |
|---|---|---|
| Img.-Val | $-0.861 / -0.905$ | $+0.929 / 0.927$ |
| Img.-R | $-0.173 / -0.172$ | $+0.928 / 0.846$ |
| Img.-S | $+0.131 / +0.148$ | $+0.915 / 0.897$ |
| Img.-A | $-0.297 / -0.257$ | $+0.910 / 0.778$ |

Table 1: **Comparing EI and JS.** Under Pearson's Correlation $r$ / Spearman's Rank Correlation $\rho$, JS does not show a strong correlation on most datasets, while EI does on all four datasets.

grayscale invariance. In Fig. 4, we find a strong linear correlation in the four cases ($r \leq 0.88$). The rank correlation is less consistent but still indicates clear trends. Moreover, most standard models have low accuracy, while models (pre)trained with more data tend to have high accuracy and invariance. We also notice standard models are scattered differently in the low-accuracy regime of ImageNet-A and ImageNet-Blur, possibly due to their dataset bias introduced during dataset construction [91–93].

## 5.4 Correlation Holds Under the CIFAR-10 Setup

We conduct a correlation study on the CIFAR-10 setup. We collect 90 CIFAR models ranging from LeNet to EfficientNet. We use the ID CIFAR-10 test set and two OOD test sets. 1) CIFAR-10.1 [94] is a reproduction of the CIFAR-10 test set but with a distribution shift arising from changes in data collection. It contains $2,000$ test images sampled from TinyImages [95]. 2) CINIC-10 test set [96] is an extended alternative for CIFAR-10. It has $90,000$ images sampled from ImageNet.

We show the relationship between model generalization and invariance in Figure 5. On all three test sets, we observe a strong correlation between model accuracy numbers and rotation invariance scores, where both $r$ and $\rho$ are greater than $0.90$. Moreover, when testing grayscale invariance, a strong correlation still exists (both $r$ and $\rho$ are greater than $0.83$).

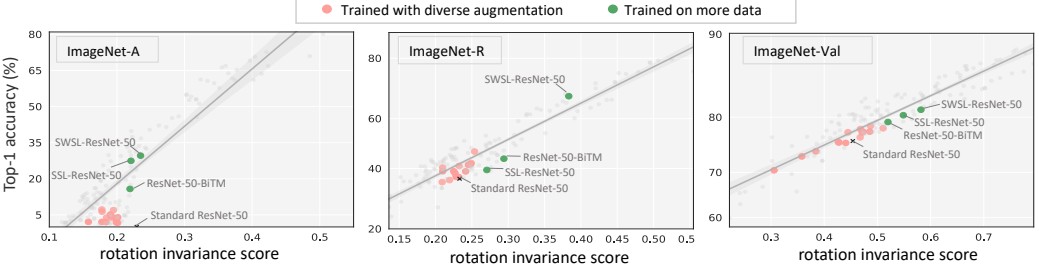

Figure 7: **Comparing data augmentation and (pre)training with more data.** We use 14 ResNet-50 models (red dots) trained with various types of augmentation methods, and 3 ResNet-50 models (green dots) trained on more data (real-world). We also mark ResNet-50 trained with the standard learning strategies. On ImageNet-Val and ImageNet-R, we observe models trained with diverse augmentation follow a linear trend. However, they deviate from the trend on ImageNet-A. In comparison, training with more data allows models to achieve relatively high accuracy and invariance on three test sets.

### 5.5 EI Gives Stronger Correlation Than JS

In the representative cases (Figure 1), EI shows superiority to JS in measuring model invariance by explicitly considering Softmax consistency and confidence. Now we compare the invariance scores measured by EI and JS *w.r.t* their correlation strength with accuracy in Figure 6 and Table 1. We find that JS provides a good correlation on the ID ImageNet-Val test set, but a much weaker correlation on the more difficult OOD tests than EI. As discussed in Section 3, the advantage of EI is that it not only follows the definition of invariance but also integrates confidence to strengthen it, while JS only compares the overall Softmax vector. In fact, the drawback of JS is primarily reflected in hard test sets, where the Softmax vector usually has a flat shape. This explains why JS does not give a strong correlation on the OOD test sets (see Section 5.3 for EI's performance on very hard test sets). We refer readers to the supplementary material for comparisons with other measures.

### 5.6 Comparing Two Training Manners of Their Generalization and Invariance

Existing works report that using more diverse training data artificially (*i.e.*, data augmentation) or naturally (*i.e.*, more real-world data) improves model invariance [4, 46–51, 97]. In this study, we compare the two strategies of their generalization and invariance abilities. We use 14 ResNet-50 models trained with strong data augmentation such as PixMix [52] and AutoAugment [48]. For comparison, we employ another 3 ResNet-50 models that learn invariance using more training samples: SWSL-ResNet-50, SSL-ResNet-50, and ResNet-50-BiTM. In Fig. 7, we observe that models with heavy and strong augmentation exhibit linear trends on ImageNet-Val and ImageNet-R, but deviate on ImageNet-A. In comparison, models (pre)trained with more data follow the linear trend on three test sets. The latter seems more effective in improving invariance and accuracy compared with data augmentation, especially on ImageNet-A. To thoroughly understand and extend this initial observation, we will conduct more comprehensive experiments on this specific point in the future.

## 6 Correlation of A Model's Invariance and Generalization on OOD Test Sets

From a **dataset-centric** perspective, we study the relationship between generalization and invariance of a given model on different OOD test sets. Given a model, we calculate its accuracy on every test set of ImageNet-C [87] and compute its rotation and grayscale invariance scores.

We evaluate ResNet-152, ViT-Base-16, and DenseNet-121 classifiers for the correlation study. As shown in Fig. 8, there is a very strong correlation between classifier accuracy and invariance on various datasets ($r > 0.93$ and $\rho > 0.95$). The results indicate that a classifier tends to have high accuracy on the test set where it has a high EI score. The above analysis indicates that it is feasible to use EI to access the out-of-distribution error of a model.

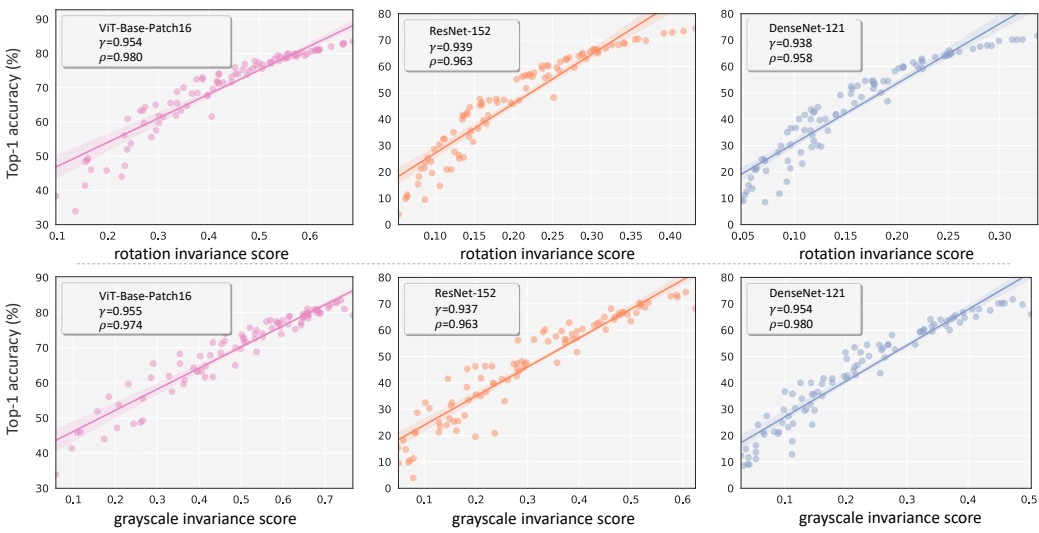

Figure 8: **Correlation between a model's invariance and accuracy on various OOD test sets.** In each figure, a data point corresponds to a test set from ImageNet-C [87]. The straight lines are fit with robust linear regression [90]. We test rotation invariance (top) and grayscale invariance (bottom). In all subfigures, we observe a strong negative correlation (Pearson's Correlation $r$ and Spearman's Rank Correlation $\rho$ are greater than $0.930$) between invariance and accuracy.

## 7    Conclusion

This work considers two critical properties of a machine learning model: invariance and generalization. To study their relationship, we first introduce effective invariance (EI) to more reasonably measure invariance and then provide an in-depth and comprehensive correlation experiment. From a model-centric perspective, we report accuracy and EI of various models have a strong linear relationship on both ID and OOD datasets, which is validated on many scenarios such as large-scale test sets, CIFAR-10, and very hard test sets. From a dataset-centric perspective, we show the accuracy and EI of a model have a strong linear relationship on various OOD datasets.

**Limitations and potential directions. First**, some networks with specially designed modules can be highly invariant to some transformations [7–9, 16, 98], such as rotation invariance networks [14, 99, 17]. The rotation invariance scores of these models may present very different correlations from our observations. That said, their color invariance scores may still exhibit a similar correlation with this work. In future works, it would be interesting to understand how these models deviate from others. **Second**, our ImageNet test sets do not include the *geographic shift* where images are captured from various locations [4]. Recent works show this shift is also a key factor influencing model accuracy [100]. We leave this question to future study. **Moreover**, we focus on classification tasks, where models are supervised by image-level annotations. In other computer vision tasks, models may be trained with different levels of supervision, such as instance-level bounding box annotations [101], pixel-level labels [102] and temporal context supervision [103]. Different types of supervision may lead to invariance ability to various factors, which may exhibit different correlation profiles. Lastly, in designing EI, we mainly study four representative cases and explain the limitations of some commonly used measures (*e.g.*, JS and L2). Considering more special cases would be beneficial.

**Potential negative social impact.** We study fundamental model properties with public classification datasets, which might be misused in certain applications with ethical concerns.

## Acknowledgments and Disclosure of Funding

We thank all anonymous reviewers for their constructive comments in improving this paper. This work was in part supported by the ARC Discovery Early Career Researcher Award (DE200101283) and the ARC Discovery Project (DP210102801). Stephen Gould is the recipient of an ARC Future Fellowship (project number FT200100421) funded by the Australian Government. Weijian Deng is a recipient of the Australian Government Research Training Program (RTP) Scholarship.

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
