# Supplementary Document for
# On the Strong Correlation Between Model Generalization and Invariance

In the supplementary document, we first compare effective invariance (EI) with other metrics *w.r.t* their correlation strength with accuracy in Section A. Moreover, we show more correlation results between a model's invariance and generalization on out-of-distribution (OOD) datasets in Section B. We also report more correlation results using ObjectNet as another ImageNet test set and introduce iWildCam-WILDS setup in Section C In addition, we show more experimental analysis in Section D. Last, we include the details of models and datasets in Section E.

## A    Comparison With Other Measures

### A.1    Comparison Under ImageNet Setup

In this section, we compare the invariance scores measured by EI and other metric *w.r.t* their correlation strength with accuracy. In the main paper, we compare EI with JS. Here, we additionally use the following five metrics.

(a) $\ell_2$ **distance** [4, 17]. It defines the invariance as the $\ell_2$ distance between predictions of original and translated data. A smaller $\ell_2$ distance indicates a higher invariance score.

(b) **Accuracy difference** [16]. The invariance is defined as the overall accuracy change on a test set after using image transformation on all test samples. A smaller difference indicates a higher invariance score.

(c) **Prediction consistency only** [2, 20]. It defines the invariance by only checking whether the predicted class is the same without considering the prediction confidence. For each test sample, if the predicted class is the same, the score is $1$; if the predicted classes are different, the score is $0$.

(d) **Prediction consistency with confidence difference** [1]. It can be viewed as a variant of EI score. For each test sample, if the predicted class is the same, then the score is defined as $1 - \|\hat{p}_t - \hat{p}\|$, where $\hat{p}$ and $\hat{p}_t$ is the prediction confidence on original data and transformed data, respectively; if the predicted classes are different, the score is $0$.

In Fig. A-1, we show the correlation results using different invariance measures. We find all measures give a clear correlation trendy on in-distribution (ID) ImageNet-Val. However, when testing on OOD datasets, $\ell_2$ distance and accuracy difference do not show a good correlation trendy. Moreover, using the model's average confidence as the invariance score cannot show high correlation strength with accuracy. Compared with measuring invariance by prediction consistency only, EI provides a relatively stronger correlation with model accuracy on OOD datasets. The above analysis shows that EI is superior to other methods in measuring model invariance by showing relatively high correlation strength with model accuracy on both ID and OOD datasets.

### A.2    Comparison Under CIFAR-10 Setup

To more comprehensively compare EI with existing measures, we newly report the correlation results under the CIFAR-10 setup. We report Spearman's rank correlation ($\rho$) between classification accuracy and rotation invariance. From the results (see Table A-1), we find when EI is used, there is a stronger correlation on three test sets under the CIFAR-10 setup. These experiments indicate that EI has very strong and stable correlations with models' OOD accuracy

36th Conference on Neural Information Processing Systems (NeurIPS 2022).

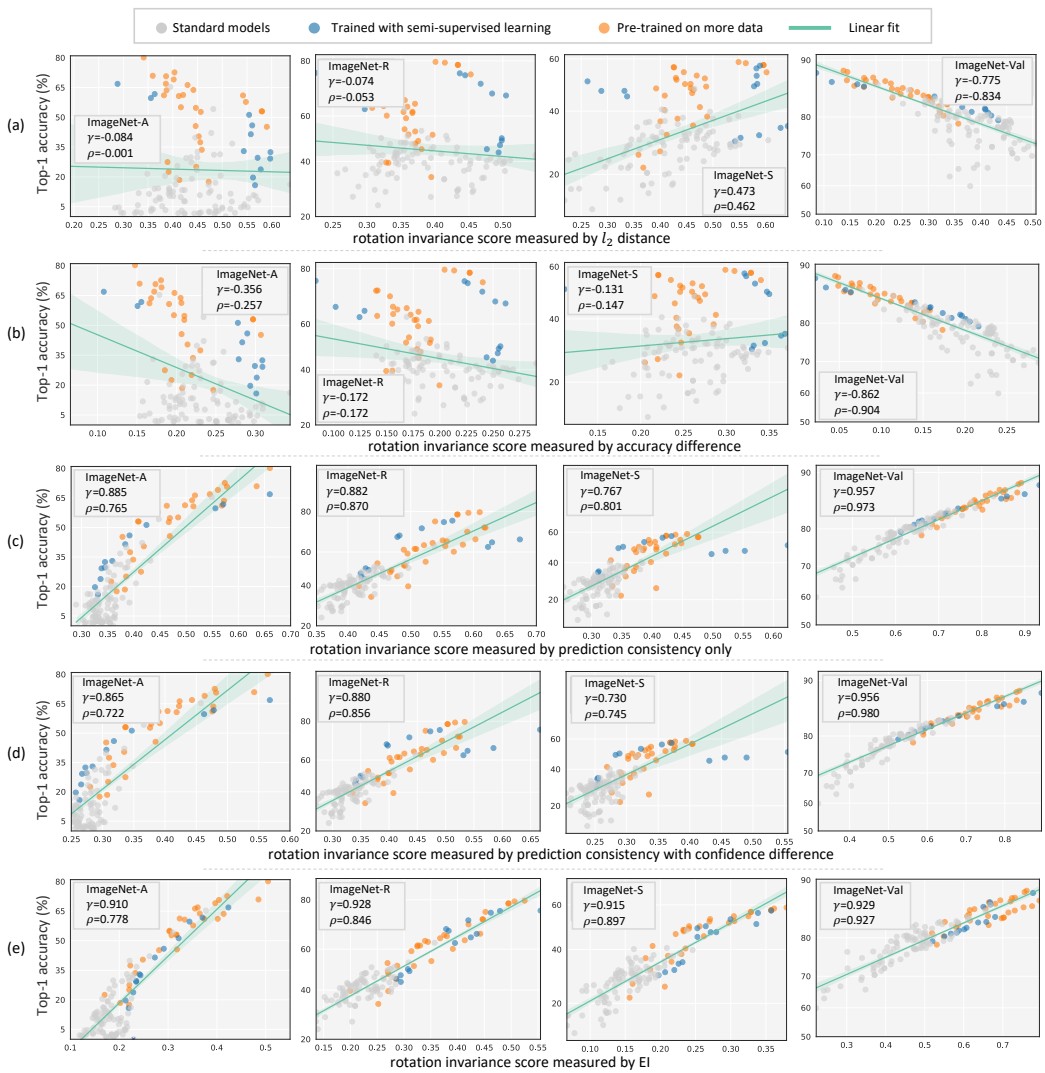

Figure A-1: **Correlation between accuracy (%) and rotation invariance.** We show the correlation results obtained using six invariance measures. From top to bottom: (a) $\ell_2$ distance, (b) accuracy difference, (c) prediction consistency only, (d) prediction consistency with confidence difference, and (e) effective invariance (EI). Compared with other measures, the invariance score measured by EI provides a reasonably good correlation with model accuracy on ID ImageNet-Val and OOD datasets (ImageNet-S/A/R).

| Test set | L2 | JS | Acc. Difference | Consistency | Consist. + Confidence | EI |
|---|---|---|---|---|---|---|
| CIFAR-10 | $-0.164$ | $+0.088$ | $+0.006$ | $+0.848$ | $+\mathbf{0.934}$ | $+\mathbf{0.934}$ |
| CIFAR10.1 | $+0.190$ | $+0.270$ | $+0.625$ | $+0.787$ | $+0.910$ | $+\mathbf{0.927}$ |
| CINIC | $+0.265$ | $+0.334$ | $+0.550$ | $+0.637$ | $+0.883$ | $+\mathbf{0.905}$ |

Table A-1: **Comparing EI and other measures under CIFAR-10 Setup.** Under Spearman's Rank Correlation $\rho$, our method still has a high correlation.

# B  More Correlation Results Between a Model's Invariance and Generalization on OOD Datasets

In this section, we additionally report the correlation results using the other three classifiers: EfficientNet-B2, Inception-V4, and RepVGG-B2. As shown in A-2, there is a very strong correlation between classifier accuracy and invariance score measured by EI on various datasets ($r > 0.900$ and

$\rho > 0.930$). This indicates that a classifier tends to have high accuracy on the test set where it has a high EI score. The above observation further shows that it is feasible to use EI score to access the out-of-distribution error of a model without using labels.

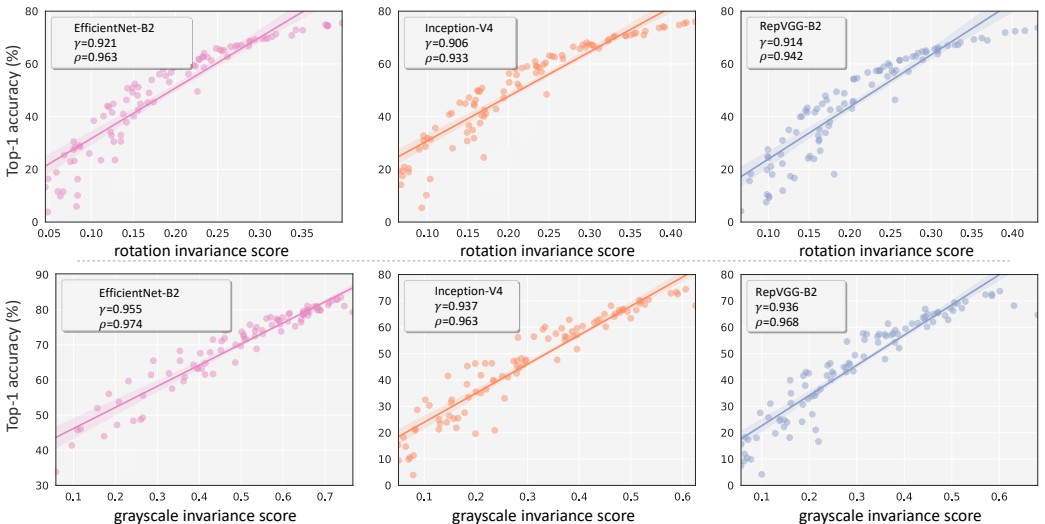

Figure A-2: **Correlation between a model's invariance and accuracy on various OOD test sets.** In each figure, a data point corresponds to a test set from ImageNet-C [7]. The straight lines are fitted with robust linear regression [10]. We test rotation invariance (top) and grayscale invariance (bottom). In each row, we use EfficientNet-B2, Inception-V4, and RepVGG-B2, respectively. In all subfigures, we observe a strong negative correlation (Pearson's Correlation $r$ and Spearman's Rank Correlation $\rho$ are greater than 0.900) between EI invariance score and model accuracy. The shaded region in each panel is a 95% confidence region for the linear fit from 1,000 bootstrap samples.

## C    More Correlation Study

### C.1    ObjectNet under ImageNet Setup

We use ObjectNet [3] as another test set under the ImageNet setup. Following the practice in [11], we evaluate classifiers on the 113 ObjectNet classes that overlap with ImageNet and report correlation study results. On ObjectNet, we observe that model invariance and generalization exhibit a strong correlation: when using rotation invariance, Pearson's Correlation ($r$) is 0.982, and Spearman's Rank correlation ($\rho$) is 0.975. Under grayscale invariance, $r$ is 0.964, and $\rho$ is 0.951.

### C.2    iWildCam-WILDS Setup

We newly use iWildCam-WILDS [11] for correlation study. iWildCam-WILDS is proposed for an animal species classification, where the distribution shift arises due to variations in camera angle, lighting, and background. Following the practice in [13], we finetune 30 classifiers that are pretrained on ImageNet using the codes provided by [11]. Then, we report the correlation results on its out-of-distribution test set (OOD Test).

We observe that under the iWildCam-WILDS setup, rotation invariance (EI) also correlates with accuracy (measured by the macro F1 score): Pearson's Correlation $r$ is 0.902, and Spearman's Rank Correlation $\rho$ is 0.915.

## D    More Experimental Analysis

**Whether using more rotation angles leads to higher correlation strength?**    To validate this, we perform a correlation study under the ImageNet setup using more randomly sampled rotations.

Specifically, we use 5 rotations (two new angles [40.1, 215.0] plus the original three 90-degree rotation angles) and 7 rotations (four new angles [38.4, 134.2, 194.0, 340.7] plus the original three 90-degree rotation angles). Results are presented in Table A-2. We observe that using more rotation angles does yield higher correlations (measured by Spearman's rank correlation).

| Test set | ImageNet-Val | ImageNet-R | ImageNet-A |
|---|---|---|---|
| 3 rotations | 0.927 | 0.846 | 0.778 |
| 5 rotation | 0.936 | 0.860 | 0.807 |
| 9 rotation | 0.948 | 0.874 | 0.814 |

Table A-2: **Correlation study using more rotation angels under CIFAR-10 Setup.** Under Spearman's Rank Correlation $\rho$, using more rotation angles does yield higher correlations.

**Consider prediction consistency in the "otherwise" case of EI.** We intuitively define the EI score as $0$ if the network gives different class predictions on the original and transformed image (the "otherwise" case in Eq.1 in the main paper). We further consider the consistency of the softmax outputs when defining the "otherwise" case in EI. Specifically, we use the negative JS in the "otherwise" case. Under this modification, the EI scores are $-0.665$ and $-0.029$ in case (c) and case (d) of Figure 1 in the main paper, respectively. Using this modified EI, we report the correlation studies on a series of benchmarks in Table A-3 (using the ImageNet models).

Compared with EI, the modified EI gives a higher Spearman's rank correlation ($\rho$) on ImageNet-Val (0.972 vs. 0.927). However, on harder OOD test sets (*e.g.*, ImageNet-S), the modified EI usually has a weaker correlation (0.422 vs. 0.897).

In designing EI, we mainly study four representative cases and explain the limitations of some commonly used measures (*e.g.*, JS and L2). We also acknowledge that considering more special cases would be beneficial for refining the "otherwise" cases of EI, but considering the complexity of this problem, this is unlikely to be achieved in the short run. As such, we would like to view the current work as a starting point that could inspire new research in the long run.

| Test set | ImageNet-Val | ImageNet-R | ImageNet-S | ImageNet-A | ObjectNet |
|---|---|---|---|---|---|
| EI | 0.927 | 0.846 | 0.897 | 0.778 | 0.975 |
| Modified EI | 0.972 | 0.764 | 0.422 | 0.575 | 0.937 |

Table A-3: **Comparing EI and Modified EI.** Modified EI only shows a strong correlation (Spearman's Rank Correlation $\rho$=0.972) on ImageNet-Val. On OOD test sets, modified EI usually has a weaker correlation than EI.

# E  Experimental Setup

## E.1  ImageNet Models

In total, we use ImageNet models provided by PyTorch Image Models (timm) [19]. They are either trained or fine-tuned on the ImageNet-1k training set [6]. The models are listed in the following.

**(1) Standard neural networks**
*{ 'resmlp_36_224', 'cait_s36_384', 'cait_s24_224', 'convit_base', 'convit_tiny', 'twins_pcpvt_base', 'eca_nfnet_l1', 'xcit_tiny_24_p8_384_dist', 'efficientnet_b1', 'efficientnet_b3', 'efficientnet_b4', 'tf_efficientnet_b2', 'tf_efficientnet_lite1', 'convnext_base', 'convnext_small', 'resnetrs350', 'pit_xs_distilled_224', 'crossvit_small_240', 'botnet26t_256', 'tinynet_e', 'tinynet_d', 'repvgg_b2g4', 'mnasnet_small', 'dla46x_c', 'lcnet_050', 'tv_resnet34', 'tv_resnet50', 'tv_resnet101', 'tv_resnet152', 'densenet121', 'inception_v4', 'resnet26d', 'mobilenetv2_140', 'hrnet_w40', 'xception', 'xception41', 'resnet18', 'resnet34', 'seresnet50', 'mobilenetv2_050', 'seresnet33ts',*

*'wide_resnet50_2', 'wide_resnet101_2', 'resnet18d', 'hrnet_w18_small', 'gluon_resnet152_v1d', 'hrnet_w48', 'hrnet_w44', 'repvgg_b2', 'densenet201', 'hrnet_w18_small', 'resnet101d', 'gluon_resnet101_v1d', 'gluon_resnet101_v1s', 'gluon_xception65', 'gluon_seresnext50_32x4d', 'gluon_senet154', 'gluon_inception_v3', 'gluon_resnet101_v1c', 'tf_inception_v3', 'tv_densenet121', 'tv_resnext50_32x4d', 'repvgg_b1g4', 'resnext26ts', 'ghostnet_100', 'crossvit_9_240', 'deit_base_patch16_384', 'rexnet_150', 'rexnet_130', 'resnetrs50', 'resnet50d', 'resnet50', 'resnetv2_50', 'resnetrs152', 'resnetrs101', 'dpn92', 'dpn98', 'dpn68', 'vgg19_bn', 'vgg16_bn', 'vgg13_bn', 'vgg11_bn', 'vgg11', 'vgg11_bn', 'vgg16', 'vgg19', 'swin_small_patch4_window7_224', 'swin_base_patch4_window12_384', 'deit_base_patch16_224', 'deit_small_distilled_patch16_224', 'densenet161', 'tf_mobilenetv3_large_075', 'inception_v3'}*

**(2) Semi-supervised learning**

*{'ssl_resnext101_32x8d', 'ssl_resnext101_32x16d', 'swsl_resnext101_32x8d', 'swsl_resnext101_32x16d', 'ssl_resnext101_32x4d', 'ssl_resnext50_32x4d', 'ssl_resnet50', 'swsl_resnext101_32x4d', 'swsl_resnext50_32x4d', 'swsl_resnet50', 'tf_efficientnet_l2_ns_475', 'tf_efficientnet_b7_ns', 'tf_efficientnet_b6_ns, 'tf_efficientnet_b4_ns, 'tf_efficientnet_b5_ns'}*

**(3) Pretraining on more data**

*{'convnext_xlarge_384_in22ft1k', 'convnext_xlarge_in22ft1k', 'convnext_large_384_in22ft1k', 'convnext_large_in22ft1k', 'convnext_base_384_in22ft1k', 'convnext_base_in22ft1k', 'resnetv2_152x2_bitm', 'resnetv2_152x4_bitm', 'resnetv2_50x1_bitm', 'resmlp_big_24_224_in22ft1k', 'resmlp_big_24_distilled_224', 'tf_efficientnetv2_s_in21ft1k', 'tf_efficientnetv2_m_in21ft1k', 'tf_efficientnetv2_l_in21ft1k', 'tf_efficientnetv2_xl_in21ft1k', 'vit_large_patch16_384', 'swin_large_patch4_window12_384', 'beit_large_patch16_512', 'beit_large_patch16_384', 'beit_large_patch16_224', 'beit_base_patch16_384', 'vit_base_patch16_384', 'vit_small_r26_s32_384', 'vit_tiny_patch16_384', 'vit_large_r50_s32_384', 'mixer_b16_224_miil', 'resmlp_big_24_224', 'resnetv2_50x1_bit_distilled', 'ig_resnext101_32x16d', 'ig_resnext101_32x32d', 'ig_resnext101_32x8d', 'ig_resnext101_32x48d', 'resnext101_32x16d_wsl', 'regnety_16gf_in1k', 'regnety_32gf_in1k', }*

**(4) ResNet-50 models trained with diverse augmentation**

(A) Models trained with augmentation policy:
'resnet50_randaa': https://github.com/zhanghang1989/Fast-AutoAug-Torch;
'resnet50_aa': https://github.com/zhanghang1989/Fast-AutoAug-Torch;
'resnet50_fastaa': https://github.com/zhanghang1989/Fast-AutoAug-Torch;

(B) Models trained with robust-related data augmentation:
'resnet50_augmix': https://github.com/google-research/augmix;
'resnet50_cutmix': https://github.com/clovaai/CutMix-PyTorch'
'resnet50_feature_cutmix': https://github.com/clovaai/CutMix-PyTorch'
'resnet50_deepaugment': https://github.com/hendrycks/imagenet-r;
'resnet50_deepaugment_and_augmix': https://github.com/hendrycks/imagenet-r;
'resnet50_pixmix': https://github.com/andyzoujm/pixmix;

(C) Adversarially "robust" models:
'resnet50_l2_eps1': https://github.com/microsoft/robust-models-transfer;
'resnet50_l2_eps0_5': https://github.com/microsoft/robust-models-transfer;
'resnet50_l2_eps0_25': https://github.com/microsoft/robust-models-transfer;
'resnet50_l2_eps0_03': https://github.com/microsoft/robust-models-transfer;
'resnet50_l2_eps0_01': https://github.com/microsoft/robust-models-transfer.

## E.2   CIFAR-10 Models

Follow the practice in [13], we train CIFAR models using the implementations from https://github.com/kuangliu/pytorch-cifar. The models span a range of manually designed architectures and the results of automated architecture searches. Specifically, we use {*'DenseNet-121/169/201/161/201', 'Densenet-cifar', 'DLA', 'DPN26/92', 'EfficientNetB0', 'GoogLeNet', 'LeNet', 'MobileNet', 'MobileNetV2', 'PNASNetA/B', 'PreActResNet18/34/50/101/152', 'RegNetX-200MF/400MF', 'RegNetY-400MF', 'ResNet-18/34/50/101/152', 'ResNeXt29-8x64d/32x4d/4x64d/2x64d', 'SENet18', 'ShuffleNetV2', 'ShuffleNetG2/G3', 'SimpleDLA', 'VGG-11/13/16/19'*}.

Furthermore, we use the trained models publicly provided by https://github.com/chenyaofo/pytorch-cifar-models. There are *{'cifar10-mobilenetv2-x0-5', 'cifar10-mobilenetv2-x0-75', 'cifar10-mobilenetv2-x1-0', 'cifar10-mobilenetv2-x1-4', 'cifar10-repvgg-a0', 'cifar10-repvgg-a1', 'cifar10-repvgg-a2', 'cifar10-resnet20', 'cifar10-resnet32', 'cifar10-resnet44', 'cifar10-resnet56', 'cifar10-shufflenetv2-x0-5', 'cifar10-shufflenetv2-x1-0', 'cifar10-shufflenetv2-x1-5', 'cifar10-shufflenetv2-x2-0', 'cifar10-vgg11-bn', 'cifar10-vgg13-bn', 'cifar10-vgg16-bn', 'cifar10-vgg19-bn'}* .

### E.3 iWildCam-WILDS Models

We use the publicly available code (https://github.com/p-lambda/wilds) to finetune models that are pretrained on ImageNet. Following the practice in [13], we with varying model architectures and learning rates. We train for 12 epochs with batch size 16 using Adam and sweep over learning rate and weight decay values in the grid $\{1e^{-3}, 1e^{-2}\} \times \{1e^{-3}, 1e^{-2}\}$ . The other Adam parameters were set to the Pytorch defaults. The models are *{ 'alexnet', 'mobilenet_v2', 'resnet18', 'resnet34', 'resnet50', 'vgg11', 'densenet121', 'regnet_y_400mf', 'shufflenet_v2_x0_5' }.*

### E.4 Datasets

The datasets we use are standard benchmarks, which are publicly available. We have double-checked their license. We list their open-source as follows.

**CIFAR-10** [12] (https://www.cs.toronto.edu/ kriz/cifar.html);
**CIFAR-10-C** [7] (https://github.com/hendrycks/robustness);
**CIFAR-10.1** [14] (https://github.com/modestyachts/CIFAR-10.1);
**CINIC** [5] (https://github.com/BayesWatch/cinic-10).

**ImageNet-Validation** [6] (https://www.image-net.org);
**ImageNet-V2-A/B/C** [15] (https://github.com/modestyachts/ImageNetV2);
**ImageNet-Corruption** [7] (https://github.com/hendrycks/robustness);
**ImageNet-Sketch** [18] (https://github.com/HaohanWang/ImageNet-Sketch);
**ImageNet-Adversarial** [9] (https://github.com/hendrycks/natural-adv-examples);
**ImageNet-Rendition** [8] (https://github.com/hendrycks/imagenet-r);
**ObjectNet** [3] (https://objectnet.dev).

### E.5 Computation Resources

We run all experiments on one 3090Ti. CPU is AMD Ryzen 9 5900X 12-Core Processor. Moreover, PyTorch version is 1.11.0+cu113 and timm version is 1.5.