# OpenReview forum: "On the Strong Correlation Between Model Invariance and Generalization"
_NeurIPS.cc/2022/Conference — NeurIPS 2022 Accept_

### Official Review · Reviewer_NZba · 2022-07-09

**Rating:** 6
**Confidence:** 3
**Soundness:** 2 fair
**Presentation:** 3 good
**Contribution:** 2 fair

**Summary:**

The authors design an empirical measure of model invariance based on prediction consistency and invariance. With this measure they evaluate the correlation between model invariance (to rotation and grayscale) and generalization on several in-distribution and out-of-distribution ImageNet variants. They show that the model’s accuracy and invariance are linearly correlated, and that such correlation is better captured by their designed measure than other existing measures.

**Questions:**

Could you also evaluate the Spearman’s rank correlation for JS-divergence on ImageNet-S and ImageNet-A? This could make the comparison fairer.

**Limitations:**

Yes, the authors addressed the limitations and potential negative societal impact of their work.

**Strengths And Weaknesses:**

### Strengths:
1. The linear relationship between model generalization and the invariance captured by the introduced measure is interesting, especially considering that the measure is handcrafted to have a specific form.

2. The evaluation covers the most recent model architectures, including several vision transformers.

### Weaknesses:
1. Current evaluation datasets are not comprehensive enough to draw convincing conclusions about the introduced measure. Most of the results are based on ImageNet variants, whereas for CIFAR-10 the authors do not do any comparison of measures. For such a handcrafted measure, readers may need similar observations on more diverse datasets, including some existing OOD benchmarks.

2. To make fair comparisons among different invariance measures, Spearman’s rank correlation is arguably a better choice than Pearson’s correlation, since different measures have different algebraic forms so a simple linear correlation may be too restrictive. I’m glad to see the authors use both correlation measures for evaluation. Nevertheless, some results are missing, including Spearman’s rank correlation for JS-divergence on ImageNet-S and ImageNet-A. Also, if we focus on Spearman’s rank correlation, the introduced measure is no longer superior in predicting generalization among other measures like the prediction consistency.

3. The motivation for designing the EI measure is subjective — why does a model making confident mistakes have the same invariance as a model making unconfident mistakes (Figure 1 (c) and (d))? If we look at the invariance from the loss perspective, the former model usually incurs a higher consistency loss than the latter one, hence less invariant. I recommend the authors to elaborate more on this motivation.

---

> ### Author Response · Authors · 2022-08-02
> **Author Response to Reviewer NZba (Part II)**
>
> **Q4: Why does a model making confident mistakes have the same invariance as a model making unconfident mistakes (Figure 1 (c) and (d))? If we look at the invariance from the loss perspective, the former model usually incurs a higher consistency loss than the latter one, hence less invariant.**
>
> Great question. We intuitively define the EI score as 0 if the network gives different class predictions on the original and transformed image (the "otherwise" case in Eq.1). Inspired by the question, we further consider the consistency of the softmax outputs when defining the "otherwise" case in EI. Specifically, we use the *negative JS* in the “otherwise” case. Under this modification, the EI scores are -0.665 and -0.029 in case (c) and case (d), respectively. Using this modified EI, we report the correlation studies on a series of benchmarks below (using the ImageNet models).
>
> |Test Set|ImageNet-Val|ImageNet-R|ImageNet-S|ImageNet-A|ObjectNet|
> |:-|:-:|:-:|:-:|:-:|:-:|
> | EI | 0.927 | **0.846** | **0.897** | **0.778**|**0.975**|
> |Modified EI|**0.972**|0.764|0.422|0.575|0.937|
>
> Compared with EI, the modified EI gives a higher Spearman’s rank correlation ($\rho$) on ImageNet-Val (0.972 vs. 0.927). However, on harder OOD test sets (*e.g.*, ImageNet-S), the modified EI usually has a weaker correlation (0.422 vs. 0.897).
>
> In designing EI, we mainly study four representative cases and explain the limitations of some commonly used measures (*e.g.*, JS and L2). We also acknowledge that considering more special cases would be beneficial for refining the "other" cases of EI, but considering the complexity of this problem, this is unlikely to be achieved in the short run. As such, we would like to view the current work as a starting point that could inspire new research in the long run.

---

> > ### Comment · Reviewer_NZba · 2022-08-09
> > **Thank you for the response**
> >
> > Most of my concerns are addressed. While the design of EI still seems unnatural to me, the empirical results are compelling. I have updated my score accordingly.

---

> > > ### Author Response · Authors · 2022-08-09
> > > **Thank you**
> > >
> > > Dear Reviewer NZba
> > >
> > > We appreciate your constructive suggestions and thank you for raising your score. We believe it would be interesting and promising to consider more special cases in the definition of invariance. We have highlighted this research point in our revised paper.
> > >
> > > Kind Regards,
> > >
> > > Authors

---

> ### Author Response · Authors · 2022-08-02
> **Author Response to Reviewer NZba (Part I)**
>
> **Q1: For CIFAR-10 the authors do not do any comparison of measures. Using more datasets for comparing invariance measures.**
>
> Good suggestion. To more comprehensively compare EI with existing measures, we newly report the correlation results under the **CIFAR-10 setup**. We report the Spearman’s rank correlation ($\rho$) between classification accuracy and rotation invariance. From the results (see table below), we find when EI is used, there is a stronger correlation on the three test sets under the CIFAR-10 setup.
>
> |Test Set|L2|JS| acc. difference| consistency|consist.+conf.|EI|
> |:-|:-:|:-:|:-:|:-:|:-:|:-:|
> |CIFAR-10|-0.164|0.088|0.006|0.848|**0.934**|**0.934**|
> |CIFAR10.1|0.190|0.270|0.625|0.787|0.910|**0.927**|
> |CINIC |0.265 |0.334|0.550|0.637|0.883|**0.905**|
>
> To further demonstrate the usefulness of EI for correlation studies, we newly use **iWildCam-WILDS** setup (see ***Q4/Reviewer G2Vw*** for more details) to compare EI with other measurements. Results are presented below. Under this setup, our method still has a high correlation ($\rho$=0.915).
>
> While correlation strength computed with "accuracy difference" is higher than EI, we emphasize that this measure gives much lower correlations under other setups. For example, under the CIFAR-10 setup, the correlation strength ($\rho$) computed by "accuracy difference" and EI is 0.625 and 0.550 on CIFAR10.1 and CINIC, respectively. These experiments indicate that EI has very strong and stable correlations with models’ OOD accuracy.
>
> |Test Set| L2|JS|acc. difference |consistency|consist.+conf.|EI|
> |:-|:-:|:-:|:-:|:-:|:-:|:-:|
> |OOD Test |0.819| 0.665 |**0.934** | 0.158 | 0.420 | 0.915|
>
> **Q2: Evaluate the Spearman’s rank correlation for JS-divergence on ImageNet-S and ImageNet-A.**
>
> Thanks for the suggestion. During rebuttal we evaluate Spearman’s rank correlation for other measures including JS-divergence on a series of benchmarks including ImageNet-S and ImageNet-A. Results are summarized in the table below.
> We observe that JS only shows a strong correlation on ImageNet-Val while EI exhibits a strong correlation in all the five datasets.
> We will update Table 1 with the additional Spearman’s rank correlation metric.
>
> |Test Set|ImageNet-Val|ImageNet-R|ImageNet-S|ImageNet-A|ObjectNet|
> |:-|:-:|:-:|:-:|:-:|:-:|
> | JS |  -0.905  | -0.172 | 0.148| -0.257 | -0.468  |
> |L2| -0.834 | -0.053 |0.462|0.001| -0.262 |
> |acc. difference| -0.904| -0.172 | -0.147 | -0.257 | 0.421 |
> |consist.| 0.973 | 0.870 | 0.801 | 0.765 | 0.955|
> |consist.+conf.| **0.980** | **0.856** | 0.745 | 0.722|0.950|
> | EI | 0.927 | 0.846 | **0.897** | **0.778** | **0.975** |
>
> **Q3: If we focus on Spearman’s rank correlation, EI is no longer superior in predicting generalization among other measures like the prediction consistency.**
>
> Great question. It is true EI gives a lower Spearman's rank correlation than a few other measures (*e.g.*, "prediction consistency") on ImageNet-S and ImageNet-Val. However, we would like to emphasize these measures lead to very unstable correlations under different setups. For example, while “prediction consistency” presents a strong correlation ($\rho$ = 0.973) on ImageNet-Val, it exhibits a much weaker correlation ($\rho$ = 0.158) on iWildCam-Wilds.
>
> In comparison, EI has strong correlations on all the test sets under the three setups (CIFAR-10, ImageNet, and iWildCam-Wilds). Thus, we think EI generally is more indicative of model generalization ability than other measures.

---

### Official Review · Reviewer_qWbK · 2022-07-11

**Rating:** 7
**Confidence:** 4
**Soundness:** 4 excellent
**Presentation:** 3 good
**Contribution:** 3 good

**Summary:**

This work considers generalization and invariance, two key properties of machine learning models. Specifically, this work conducts a large-scale quantitative correlation study between generalization and invariance on ImageNet and CIFAR setups. To help the correlation study, this work introduces the Effective Invariance Score (EI). It is more suitable for measuring model invariance than other measures such as KL and L2. Two key observations are reported: 1) generalization and invariance exhibit strong linear relationships on both in-distribution and out-of-distribution test sets; 2) when tested on different test sets, the accuracy of the model is linearly related to its invariance ability.

**Questions:**

The following questions and suggestions are expected to be addressed in the rebuttal.

[1] First, lines 97-105 explain why KL is not used for correlation studies. I understand why KL and other measures are not suitable. However, using "Drawback of existing invariance measures" to summarize the paragraph is not very coherent and is a bit confusing.

[2] Also, in semi-supervised learning, other additional techniques such as confidence [48] and sharpness [49] are used when KL [50, 51] or L2 are used for maintaining consistency. These techniques may avoid the problems illustrated in case (c) and case (d) in Figure 1. It can also be mentioned on lines 97-105 to help illustrate why KL and L2 are not suitable for correlation studies at work.

[3] As illustrated in the weaknesses, a correlation study on ObjectNet would be interesting.

[4] In addition, related works (eg, [a,b]) should be included in this work. Section 5.7 should be highlighted (see Weaknesses above).


**Ethics Review Area:**

["I don’t know"]

**Limitations:**

In Section 6, the authors clearly discuss potential limitations and propose possible solutions and potential research directions. Potential negative social impacts are also discussed.

**Strengths And Weaknesses:**

[Strengths]
[1] Overall, the work is well written. Motivation, focus, experimental setup, and key observations are clearly presented. Related work is well organized.

[2] The proposed Invariance Score (EI) is reasonable and well-discussed. The authors give an intuitive illustration in Figure 1. Furthermore, comparisons with other measures in the Supplementary file help to understand why EI is more suitable for correlation studies.

[3] Existing work mainly gives qualitative hints on the relationship between invariance and generalization. The main contribution of this work is it provides quantitative analysis. Furthermore, observations are presented on both in-distribution and out-of-distribution test sets. This is another contribution compared with the existing works. The results on the ImageNet testbed are comprehensive and systematic.

[4] Another interesting observation is that model invariance and accuracy also exhibit strong linear correlations on various test sets (Section 5.7). The results are reported on ImageNet-C with several models, which are helpful.

[5] In addition to the two main observations, this work investigates other points, including how the correlation on the hard test set varies (Section 5.3), and why JS is not suitable for correlation studies (Section 5.5 and in Supplementary Material). These points help to understand the relationship between model invariance and generalization.

[Weaknesses]
[1] This work quantitatively investigates the relationship between model invariance and generalization. This work points out that previous work [11-13, 15, 28] gave some qualitative hints about the positive correlation between them (Lines 23-29, 68-75). However, some recent works such as [a] and [b] are missing. They should be included and discussed.

[a] Zhu Sicheng, Bang An, Huang Furong. "Understanding the generalization advantage of model invariance from a data perspective." Advances in Neural Information Processing Systems 34 (2021): 4328-4341.
[b] Sannei, Akiyoshi, Masaki Imaizumi, and Makoto Kawano. "Improving generalization bounds for group-invariant/equivariant deep networks through quotient feature spaces." Uncertainty in Artificial Intelligence. PMLR, 2021.

[2] Continuing the above point, existing works mainly focus on the in-distribution generalization. In contrast, this work investigates both in- and out-of-distribution test sets. Differences or new contributions should be further highlighted in Section 2. The current version (lines 68-75) is not very clear.

[3] This work conducts correlation studies on the ImageNet testbed, which includes several test sets. These test sets are diverse and come from different distributions (Lines 154-157). It is good and convincing. Another test set is ObjectNet, where object backgrounds, rotations, and viewpoints are diverse. It would be better if this work could include this test set in the correlation study.

[4] Section 5.7 should be highlighted as it presents another key observation. However, it is not heightened in the current version. In addition, the results in the Supplementary Materials (Figure A-3) should be included in the main paper, which helps illustrate the observations.

---

> ### Author Response · Authors · 2022-08-02
> **Author Response to Reviewer qWbK**
>
> **Q1: Using "Drawback of existing invariance measures" to summarize the paragraph (lines 97-105) is not very coherent and is a bit confusing.**
>
> A: Thank you for pointing this out. To avoid confusion, we will use “limitations of some commonly used invariance measures”.
>
> **Q2: Additional techniques such as confidence [48] and sharpness [49] are used when KL [50, 51] or L2 are used for maintaining consistency. It can also be mentioned these techniques to help illustrate why KL and L2 are not suitable for correlation study.**
>
> A: Good idea. We fully agree that these techniques (confidence [48] and sharpness [49]) can be used to discard samples (e.g., case d) with flat softmax outputs when computing the consistency loss. This will avoid scenarios where KL and L2 mistake them as samples with high prediction consistency. We will discuss these techniques in Lines 103-105.
>
> **Q3: Correlation study on ObjectNet would be interesting.**
>
> A: Great suggestion. Following the practice in [2], we evaluate classifiers on the 113 ObjectNet classes that overlap with ImageNet and report correlation study results.
>
> |Type|Pearson’s Correlation ($r$)|Spearman’s Rank Correlation ($\rho$)|
> |-----|:------:|:------:|
> |Rotation | 0.982 | 0.975|
> |Grayscale | 0.964 | 0.951|
>
> ***On ObjectNet, we observe that model invariance and generalization exhibit a strong correlation***: when using rotation invariance, Pearson's correlation ($r$) is 0.982, and Spearman’s Rank correlation ($\rho$) is 0.975. Under grayscale invariance, $r$ is 0.964, and $\rho$ is 0.951.
>
> [2] Miller, John P., et al. "Accuracy on the line: on the strong correlation between out-of-distribution and in-distribution generalization." International Conference on Machine Learning. PMLR, 2021.
>
> **Q4: Related works ( [a,b]) should be included.**
>
> A: Thanks. Work [a] derives model complexity bounds based on the sample cover induced by data transformations. Work [b] develops a new generalization error bound using the proposed quotient feature space for invariant and equivariant deep neural networks. Both works qualitatively suggest that learning invariant features benefits generalization. Different from them, we perform a large-scale quantitative correlation study using various models and different test sets and empirically report that model invariance and generalization exhibit a strong linear correlation on both in- and out-of-distribution test sets. We will update Lines 23-29 to discuss both works.
>
> *[a] Zhu Sicheng, Bang An, Huang Furong. "Understanding the generalization advantage of model invariance from a data perspective." In NeurIPS, 2021*
>
> *[b] Sannei, Akiyoshi, Masaki Imaizumi, and Makoto Kawano. "Improving generalization bounds for group-invariant/equivariant deep networks through quotient feature spaces." Uncertainty in Artificial Intelligence, 2021*
>
>
> **Q5: Differences or new contributions should be further highlighted in Section 2.**
>
> A: Thanks. Lines 59-75 of Section 2 introduce works of predicting generalization in deep learning (PGDL). These works generally assume an in-distribution test set and study limited types of networks. In comparison, we conduct a much more comprehensive study on both in-distribution and out-of-distribution test sets, using various network architectures. We will update Lines 68-75 to highlight the difference between our work and PDGL.
>
>
> **Q6: Section 5.7 should be highlighted as it presents another key observation.**
>
> Thanks. We will make Section 5.7 a separate section and integrate the correlation results (Figure A-3) in the main paper.

---

> > ### Comment · Reviewer_qWbK · 2022-08-08
> > **Thanks for your response**
> >
> > Rebuttal has solved all my concerns. I will maintain my initial score.

---

> > > ### Author Response · Authors · 2022-08-08
> > > **Thank you**
> > >
> > > Dear Reviewer qWbK,
> > >
> > > Thank you for your valuable comments and constructive suggestions! We are happy to hear that your concerns have been resolved.
> > >
> > > Kind Regards,
> > >
> > > Authors

---

### Official Review · Reviewer_G2Vw · 2022-07-11

**Rating:** 7
**Confidence:** 5
**Soundness:** 4 excellent
**Presentation:** 4 excellent
**Contribution:** 3 good

**Summary:**

This work proposes a very simple to implement, yet effective, metric (effective invariance or EI) to assess the invariance of a model with respect to some input transformation. The main novelty of the proposed method is that it does not rely on the true label, but rather on the agreement between the predictions given an image and its transformed version.

The main findings of the paper are about the role of invariance in multi-class classification problems. The authors find that invariance (as measured by EI) is strongly correlated with accuracy for a large set of models and datasets. Moreover, it is empirically proven that EI is much better than Jensen-Shannon Divergence at capturing invariance for OOD datasets, showing that EI is a more robust metric for invariance.

An additional (and important) observation included in the paper is that training with additional data is more effective (in terms of invariance) than any of the augmentation strategies tried.

The experimental setup is very solid, with a large breadth of models and datasets being evaluated. Moreover, the paper is well written and the language used is clear and concise.

**Questions:**

* DNNs are known to be very sensitive to specific translations [1] even if they are "hard-wired" to be translation invariant. This sensitiveness is much stronger for other transformations for which the DNN has learnt the invariance. Could you comment on the use of three 90 degree rotations per sample to compute EI. Could this be missing many cases where the model is invariant (or not)? I believe EI could be used to "map" the invariance of a CNN in the full transformation domain, which could be extremely important to guide machine learning practitioners to improve their models.

* I would suggest including randomly initialized networks (not trained) in the study. These networks could also be invariant, but do they follow the same trend shown in the plots? Or training till convergence is a condition for EI to work? The study of randomly initialized networks could also shed some light on the invariance induced by the architectural design alone.

* What is the correlation between EI(grayscale) and EI(rotation). Intuitively, they are strongly correlated; if so, have the authors thought about understanding which of the two is the most important source of invariance for generalization?


[1] Biscione and Bowers, Convolutional Neural Networks Are Not Invariant to Translation, but They Can Learn to Be, JMLR 2021.

**Limitations:**

I believe the authors have addressed the limitations of their work correctly, pointing out:
* The scope of the paper is on multi-class classification. The study (and maybe new formulations of EI) could be extended to regression, detection, etc.
* The inherent invariances in some architectures (eg. hard-wired rotation invariance), which are not included in this study.
* The impact of *geographic shift*, not included in this work.

Additionally, the authors comment on the societal impact, which is limited since this work uses already pre-trained models. The major impact is inherited from the impact of each of these models themselves.

**Strengths And Weaknesses:**

**Strengths:**

* The proposed measure for invariance is extremely simple to implement, and well backed up by examples and experimental results.
* The experimental setup is very solid, with a large amount of state of the art models being analyzed. The distinction between fully-supervised, semi-supervised and models trained with more data is smart, and allowed the authors to reach some conclusions in terms of the use of more data in training wrt. augmentations.
The comparison with other measures for invariance in Fig. 6, Table 1 and Appendix A is very interesting, and also clearly favorable for EI.
* The paper's language is clear, I enjoyed the reading. The experimental setup is very well described, with links to all models used and datasets. I believe one of the major strengths of this paper is the experimental setup itself.

**Weaknesses:**

* Although the metric proposed is evaluated using 3 rotations of 90 degrees per image, I believe the method could benefit from using multiple transformed versions of $x$ (eg. randomly sampled rotations), and then measuring the agreement similarly to Eq. (1). However, I believe the proposed 3-way rotation is enough to prove the validity of EI in this work.

* The EI proposed in Eq. (1) is always 0 when there is a disagreement between the prediction of the unaugmented $x$ and the augmented $x^\prime$. It is reasonable to discard all cases where both predictions disagree, but many times they might mildly disagree, even within the expected standard deviation. I think that the "otherwise" case in Eq. (1) could be further refined accounting for those cases where it is not clear whether the model is invariant or not.

* All the OOD datasets analyzed are modifications of Imagenet, which was used for training. I really appreciated the breadth of OOD datasets tested, but I missed at least one dataset that is strongly OOD, for example train on Imagenet and finetune for a dataset that is completely unrelated to Imagenet. I expect EI to perform well in those cases too, so please take this as a suggestion to further improve the paper.

---

> ### Author Response · Authors · 2022-08-02
> **Author Response to Reviewer G2Vw (Part II)**
>
>
> **Q5: I would suggest including randomly initialized networks (not trained) in the study. These networks could also be invariant, but do they follow the same trend shown in the plots? Or training till convergence is a condition for EI to work? The study of randomly initialized networks could also shed some light on the invariance induced by the architectural design alone.**
>
> Interesting suggestion. First, during rebuttal we tested 20 randomly initialized networks (e.g., ViT and EfficientNet) on 7 ImageNet test sets. We observe that both their rotation invariance and classification accuracy *are very low* and cluttered (no linear relationship) and conclude that these networks do not have good invariance capacity. So for EI to work we cannot use randomly initialized models.
>
> Furthermore, we also tested 10 CIFAR-10 classifiers that are not trained till convergence. Specifically, the classifiers are only trained with a few epochs. We observe that they still follow a similar linear trend. Therefore, training till convergence is not a necessary condition for EI to work.
>
> **Q6: What is the correlation between EI (grayscale) and EI (rotation). Intuitively, they are strongly correlated; if so, which of the two is the most important source of invariance for generalization?**
>
> Interesting point. We observe that they are indeed strongly correlated. In our new experiment, the Spearman's rank correlation $\rho$ is 0.947, 0.950, and 0.965 on ImageNet-Val, ImageNet-S, ImageNet-R, respectively. It suggests that the network simultaneously gains rotation and grayscale invariance.
>
> Regarding which invariance is more important for generalization, our correlation studies (Figures 2 and 3) show that rotation invariance generally has a stronger correlation with accuracy than grayscale invariance (5 out of 6 test sets). The only case for grayscale to have a stronger correlation is ImageNet-R, which is featured by style shift. We think under style shift, the model probably has more incentives to be invariant to color changes.
>
> In the real world, images often exhibit diverse geometric and color variations. To measure generalization in these scenarios, we think both rotation and grayscale invariance are critical.

---

> > ### Comment · Reviewer_G2Vw · 2022-08-04
> > **Good rebuttal**
> >
> > I thank the authors for a clear and concise rebuttal, where the main issues raised in the review were addressed. I believe the paper is of great interest, particularly the thorough evaluation, which is even more complete after rebuttal.
> > For these reasons, and those mentioned in my review, I maintain my initial score.

---

> > > ### Author Response · Authors · 2022-08-04
> > > **Thank you**
> > >
> > > Dear Reviewer G2Vw,
> > >
> > > Thank you for your positive assessment and constructive feedback on our work. We gratefully acknowledge that your suggestions helped us to improve our manuscript.
> > >
> > > Best,
> > >
> > > Authors

---

> ### Author Response · Authors · 2022-08-02
> **Author Response to Reviewer G2Vw (Part I)**
>
> **Q1: The method could benefit from using multiple transformed versions of x (eg. randomly sampled rotations). However, the proposed 3-way rotation is enough to prove the validity of EI in this work.**
>
> Good idea. During rebuttal, we performed a correlation study under the ImageNet setup using more randomly sampled rotations. Specifically, we use 5 rotations (two new angles [40.1, 215.0] plus the original three 90-degree rotation angles) and 7 rotations (four new angles [38.4, 134.2, 194.0, 340.7] plus the original three 90-degree rotation angles).  Results are presented below.
>
> |Test Set|ImageNet-Val|ImageNet-R|ImageNet-A|
> |:-|:-:|:-:|:-:|
> | 3 rotations | 0.927 | 0.846 | 0.778 |
> | 5 rotations | 0.936 | 0.860 | 0.807|
> | 7 rotations | **0.948** |**0.874**| **0.814**|
>
> We observe that using more rotation angles does yield higher correlations (measured by Spearman's rank correlation). We will add these results in the revised version.
>
> **Q2: Could you comment on the use of three 90 degree rotations per sample to compute EI. Could this be missing many cases where the model is invariant (or not)? I believe EI could be used to "map" the invariance of a CNN in the full transformation domain, which could be extremely important to guide machine learning practitioners to improve their models.**
>
> Insightful idea. We would like to share our thoughts from the following aspects. First, for rotation invariance, using three 90-degree angles satisfies our basic needs, as illustrated in the experiment. During rebuttal, we further find that using more rotation angles is beneficial (see our reply to Q1). It likely means using more angles captures finer details of a model’s invariance property.
>
> Second, rotation invariance measured in our work may not be sufficient to reflect invariance to other transformations (*e.g.*, shear and illumination change). If we could analyze the invariance of a CNN in the full transformation domain, we would probably be able to gain a more comprehensive understanding of model generalization / invariance capacities.
>
> **Q3: It is reasonable to discard all cases where both predictions disagree, but many times they might mildly disagree, even within the expected standard deviation. The "otherwise" case in EI (Eq. 1) could be further refined accounting for those cases.**
>
> A: Good suggestion. In our work, we mainly discuss four representative cases and explain the limitations of existing measures (*e.g.*, JS and L2) for correlation studies. In light of your question, during rebuttal we modify the proposed EI to consider predictions that "slightly disagree". More details can be viewed in our response to ***Q4/Reviewer NZba***.
>
> We show that the modified EI gives a stronger correlation only on ImageNet-Val and a weaker correlation on other test sets. From this trial, we think it is not easy to improve EI that covers some unconsidered cases, especially in this short period. The proposed method can thus serve as a starting point to inspire new research in the long run.
>
> **Q4: I missed at least one dataset that is strongly OOD: train on Imagenet and finetune for a dataset that is completely unrelated to Imagenet. Please take this as a suggestion to further improve the paper.**
>
> Good suggestion. In the light of this suggestion, we newly use iWildCam-WILDS [a] for correlation study during rebuttal. iWildCam-WILDS is an animal species classification dataset, where the distribution shift arises due to variations in camera angle, lighting, and background.
>
> *[a] Koh, P. W., Sagawa, S., Marklund, H., Xie, S. M., Zhang, M., Balsubramani, A., Hu, W., Yasunaga, M., Phillips, R. L., Beery, S., et al. WILDS: A benchmark of in-the-wild distribution shifts. In ICML, 2021*
>
> Following the practice in [3], we finetune 30 classifiers that are pretrained on ImageNet using the codes provided by [a]. Then, we report the correlation results on its out-of-distribution test set (OOD Test).
> We show that under the iWildCam-WILDS setup, rotation invariance (EI) also correlates with accuracy (measured by the macro F1 score). Please refer to the table below.
>
> |Test Set|Pearson’s Correlation ($r$)|Spearman’s Rank Correlation ($\rho$)|
> |-----|:------:|:------:|
> |OOD Test | 0.902 | 0.915|

---

### Official Review · Reviewer_h3Wy · 2022-07-12

**Rating:** 2
**Confidence:** 4
**Soundness:** 1 poor
**Presentation:** 2 fair
**Contribution:** 1 poor

**Summary:**

The paper starts with three claims in existing works that motivate their work---(1) the relationship between generalization and invariance are largely qualitative in existing work, (2) they are restricted to in-distribution datasets, and that (3) existing quantitative methodologies to quantify invariance do not take into account confidence of predictions. To address these, the authors propose a metric for invariance (EI), which uses both consistency and confidence of model predictions. Then, they conduct an evaluation of how generalization and invariance are correlated across a variety of architectures.

**Questions:**

N/A

**Limitations:**

Yes.

**Strengths And Weaknesses:**

Strengths:

1. The problem is extremely important. Understanding the mechanisms driving generalization is a fundamental problem in ML. Invariance has been proposed as one potential mechanism generalization could be achieved. And so, studying their relationship in deep models is important.

2. Scale of the experiments conducted is large.

Weaknesses:

1. Severely incorrect claims: Unfortunately, there are serious flaws at in the claims made in this work. Below I list them with reasoning/citations:

a) Existing works are qualitative: There is a very long list of existing works which study this relationship quantitatively, and have reported it before [1-8]
b) Existing works are in-distribution: In [7,8,2] authors have studied specifically the problem of out-of-distribution and invariance. In fact, [2] presents new, complex datasets to study specifically this particular problem.
c) Not taking into account confidence: Invariance has been long measured in neuroscience with metrics that take into account confidence. Adaptations of this metric can be seen in [1,2,3,4]. Theoretical investigations [7,8] have also presented much more articulate representations of invariance measurement.
d) No strong quantitative correlation shown on OOD data: Both [1,2] show quantitatively that generalization positively correlates with invariance. This has also been discussed at length in previous works including [7,8].

2. Completely missing a large portion of literature on invariance and generalization: There is a large body of literature on invariance and generalization (beyond the ones cited below), which this paper has completely ignored. It is not placed in the context of these works, and that may be partly why the paper has made incorrect claims due to being unaware of existing works.

3. No novel contribution: The formulation of EI is very similar to those proposed in [1], and the specialization score presented in [2]. It is what EI adds beyond existing work, as the problems mentioned in this paper as motivations are not true, and have been already addressed by past work which the paper has not referred to.

In summary: This paper seems unaware of a large subfield of existing literature in neuroscience and AI, which has led to incorrect claims about existing works. It is unclear how EI adds beyond existing works. All in all, the paper needs a major rethinking and reevaluation of the contributions in light of a thorough literature review on generalization and invariance.

References

1. https://www.nature.com/articles/s41593-018-0310-2
2. https://www.nature.com/articles/s42256-021-00437-5
3. https://papers.nips.cc/paper/2009/file/428fca9bc1921c25c5121f9da7815cde-Paper.pdf
4. https://papers.nips.cc/paper/1997/file/792c7b5aae4a79e78aaeda80516ae2ac-Paper.pdf
5. https://arxiv.org/pdf/1706.01350.pdf
6. https://papers.nips.cc/paper/1996/hash/2812e5cf6d8f21d69c91dddeefb792a7-Abstract.html
7. Poggio, T.A. and Anselmi, F., 2016. Visual cortex and deep networks: learning invariant representations. MIT Press.
8. https://academic.oup.com/imaiai/article-abstract/5/2/134/2363483?redirectedFrom=fulltext

---

> ### Author Response · Authors · 2022-08-02
> **Author Response to Reviewer h3Wy (Part II)**
>
> **Q5: Authors incorrectly claim “existing works are in-distribution”, but “In [7,8,2] authors have studied specifically the problem of out-of-distribution and invariance. In fact, [2] presents new, complex datasets to study specifically this particular problem.”**
>
> Thanks for this comment. We would like to point out that ***[7,8] does not*** use out-of-distribution data or study the relationship of OOD and invariance: they only study invariance. On the other hand, ***[2] studies a very specific OOD problem***: recognizing objects under new viewpoints, while the OOD generalization we study is much more general (much more beyond viewpoint variation) and widely acknowledged in the community [b,c]. Further, as mentioned in our reply to *the above Q4*, ***[2] does not study pure invariance vs. OOD***, but a mixed property, and this investigation is on the neuron level instead of the network level as we do. In addition, the scale of the experiments in [2] is limited: it only studies a few network structures.
>
> *[b] Taori, Rohan, et al. "Measuring robustness to natural distribution shifts in image classification." In NeurIPS, 2020*
>
> *[c] Hendrycks, Dan, et al. "The many faces of robustness: A critical analysis of out-of-distribution generalization." In ICCV, 2021*
>
>
> **Q6: Authors incorrectly claim “no strong quantitative correlation shown on OOD data”, but “both [1,2] show quantitatively that generalization positively correlates with invariance. This has also been discussed at length in previous works including [7,8]”**
>
> Thanks for the comment. We find that ***OOD generalization is not discussed in the studies of [1,7,8]***, so there is no report of the quantitative relationship between OOD and invariance of [1,7,8]. In fact, there is even no mention of invariance in [1].
>
> Regarding [2], it only studies the relationship between specialization score and a specific OOD problem (*i.e.*, unseen category-viewpoint combination). As mentioned above, the specialization score ***is not equal to*** the invariance score, and this specific OOD problem ***is not representative*** of the general OOD generalization problem the machine learning community is studying.
>
> As such, we feel that there is no obvious error in our claim.
>
> **Reference**
>
> *[1] Yang, Guangyu Robert, et al. "Task representations in neural networks trained to perform many cognitive tasks." Nature neuroscience, 2019*
>
> *[2] Madan, Spandan, et al. "When and how convolutional neural networks generalize to out-of-distribution category–viewpoint combinations." Nature Machine Intelligence, 2022*
>
> *[3] Goodfellow, Ian, et al. "Measuring invariances in deep networks." In NIPS, 2009*
>
> *[4] Riesenhuber, Maximilian, and Tomaso Poggio. "Just one view: Invariances in inferotemporal cell tuning." In NIPS, 1997*
>
> *[5] Achille, Alessandro, and Stefano Soatto. "Emergence of invariance and disentanglement in deep representations." The Journal of Machine Learning Research, 2018*
>
> *[6] Bricolo, Emanuela, Tomaso Poggio, and Nikos K. Logothetis. "3D object recognition: A model of view-tuned neurons." In NIPS, 1996*
>
> *[7] Poggio, Tomaso A., and Fabio Anselmi. Visual cortex and deep networks: learning invariant representations. MIT Press, 2016*
>
> *[8] Anselmi, Fabio, Lorenzo Rosasco, and Tomaso Poggio. "On invariance and selectivity in representation learning." Information and Inference: A Journal of the IMA, 2016*

---

> ### Author Response · Authors · 2022-08-02
> **Author Response to Reviewer h3Wy (Part I)**
>
>
> **Q1: Authors incorrectly claim “existing works are qualitative”, but “there is a very long list of existing works which study this relationship quantitatively” [1-8].**
>
> Thank you for pointing out these works, which we carefully read during rebuttal. We find there is no quantitative analysis of the relationship between invariance and generalization: these works only quantitatively study invariance itself. We therefore believe our claim is correct.
>
> **Q2: Authors incorrectly claim existing works are “not taking into account confidence”, but “Invariance has been long measured in neuroscience with metrics that take into account confidence.” [1-4].**
>
> We respectfully disagree. The suggested papers ***do not really discuss the confidence*** as we know it in deep neural networks [a]. Specifically, the suggested papers [1-4] study ***response values of an individual neuron*** for the input stimulus, where such response values are at the local neuron level. In comparison, “confidence” in our work refers to ***class prediction score*** given by the classifier for an input image, and such confidence value is on the global network level. This definition of confidence is widely adopted in the community [a]. Therefore, we believe our claim is correct.
>
> *[a] Goodfellow, Ian, Yoshua Bengio, and Aaron Courville. Deep learning. MIT press, 2016.*
>
> **Q3. Theoretical investigations [7,8] have also presented much more articulate representations of invariance measurement.**
>
> Great comment. We agree both works give interesting insights of invariance measurement on the ***representation level***. Specifically, they define that a representation map is invariant if it gives the same representation on the original and transformed data.
> However, this representation-level invariance measurement ***cannot be applied*** under this paper’s context: analyzing different models of their invariance vs. generalization ability. The reason is that different models have very different representations (*e.g.*, dimensionality). In comparison, EI does not work on the feature level: the proposed usage of confidence score allows us to compare different models effectively.
>
> **Q4: No novel contribution: The formulation of EI is very similar to those proposed in [1], and the specialization score presented in [2].
> Thank you for this comment. We would like to emphasize that EI is very different from fractional task variance (FTV) [1] and specialization score [2].**
>
> First, ***FTV [1] does not*** measure invariance. Instead, FTV describes which tasks a neuron tends to learn.
>
> Second, ***specialization score [2]*** measures a neuron’s certain property which is a ***combination*** of selectivity and invariance instead of invariance itself. Specifically, in its formula, the specialization score is computed by an invariance score and a selectivity score: the latter two have different physical meanings, and their geometric mean no longer measures invariance itself.
>
>
> Third, the invariance score considered in [2] is computed on the ***neuron level***: the invariance score of a neuron is calculated according to its response/activity changes on the transformed stimulus. In comparison, our EI is on the ***network level***: we measure how consistent the network's class predictions are on transformed test data.
>
>
> We will add the above discussions to the revised paper.

---

> ### Author Response · Authors · 2022-08-09
> **Response Follow-Up**
>
> Dear Reviewer h3Wy,
>
> Thank you for acknowledging that our research problem is *"extremely important"* and *"scale of the experiments conducted is large"*.
> We also thank you for pointing out the computational neuroscience papers. After reading them carefully, we find that they *do not* perform large-scale quantitative analysis as we do to show the relationship between invariance and network performance. In our revised paper, we have included these works and discussed their differences from our work (Lines 23-31 in the main paper).
>
> Furthermore, we have highlighted the essential difference between our network-level EI and neuron-level invariance measures (please see Q2-Q4 in the response and Lines 110-111 in the main paper).
>
> In addition, we have clarified that the relationship between out-of-distribution generalization and invariance is *not* discussed in referenced works (please see Q5 and Q6).
>
> We believe ***our claims are correct in the context of the referenced works***. We hope our response has addressed your initial concerns. Please let us know if you have any other questions.
>
> Best,
>
> Authors

---

### Author Response · Authors · 2022-08-07
**Common Response**

Dear Reviewers,

Thank you for your detailed and thoughtful feedback. Inspired by your valuable suggestions, we have added more experimental analyses and included the suggested related works. We have also updated the main paper and supplementary material. We summarize the major changes below:

- **Correlation study under the iWildCam-WILDS setup**. We show that under this new setup, rotational invariance (EI) is also correlated with accuracy (*Q4/Reviewer G2Vw, Section D.2 in the Supp.*).


- **More research on EI**. ***First***, we compare our EI with existing measures (*e.g.*, JS and L2) on the CIFAR-10 and iWildCam-WILDS setups (*Q1/Reviewer NZba, Section A.2 in the Supp.*). We show that other measures lead to very unstable correlations under different setups. In contrast, EI strongly correlates with accuracy on all test sets under the three setups. ***Second***, inspired by the suggestions of Reviewers NZba and G2Vw, we discuss a modified version of EI that considers the prediction consistency in the "otherwise" case of EI. We show that EI is more stable than this modified version on the ImageNet setup (*Q4/Reviewer NZba, Section E in the Supp.*).

- **More experimental analyses**. ***First***, we show that model invariance and generalization exhibit a strong correlation on ObjectNet (*Q4/Reviewer qWbK, Section D.1 in Supp.*).
 ***Second***, we show that using more rotation angles to measure rotation invariance does show higher correlations (*Q1/Reviewer G2Vw, Section E in Supp.*). ***Third***, we show that the accuracy and invariance of randomly initialized models are very low and cluttered (no linear relationship) (*Q5/Reviewer G2Vw*).

- **More related works**. ***First***, we include computational neuroscience works (*suggested by Reviewer h3Wy*). ***Second***, two theoretical studies (*suggested by Reviewer qWbK*) are also included. We also highlight our work's differences (*Lines 23-28 in the main paper*).

We hope our response has addressed the initial concerns. Please let us know if you have any other questions.

Kind Regards,

Authors

---

### Meta-Review · Area_Chair_bzKy · 2022-08-27

**Recommendation:** Accept
**Confidence:** Less certain

**Metareview:**

This work proposes a very simple to implement, yet effective, metric (effective invariance or EI) to assess the invariance of a model with respect to some input transformation. The main novelty of the proposed method is that it does not rely on the true label, but rather on the agreement between the predictions given an image and its transformed version.

The committee appreciates the comprehensiveness of the empirical evaluation conducted in the paper. Although theoretical analysis on how invariances improves generalization exists, as pointed out by the reviewers (missed in the paper's initial version), this paper performs a large-scale quantitative correlation study using various models and different test sets and empirically reports that model invariance and generalization exhibit a strong linear correlation on both in- and out-of-distribution test sets.

There is a heated discussion about the novelty of this work, as one reviewer pointed out that the authors missed a large subfield of existing literature in neuroscience and AI. The rest of the committee, however, does recognize the differences and believes that the efforts of the large scale experimental evaluations outweigh. The authors, however, are strongly suggested to provide a detailed comparison of the mentioned works in the revised paper.

**Award:**

No

---

### Decision · Program_Chairs · 2022-09-14

Accept